# A Time Bank System Design on the Basis of Hyperledger Fabric Blockchain

**Yu-Tse Lee [1], Jhan-Jia Lin [1], Jane Yung-Jen Hsu [2] and Ja-Ling Wu [2,\*]**

[1]  Department of Electrical Engineering, National Taiwan University, Taipei 106, Taiwan;
    r05921105@ntu.edu.tw (Y.-T.L.); r05921060@ntu.edu.tw (J.-J.L.)
[2]  Department of Computer Science and Information Engineering, National Taiwan University,
    Taipei 106, Taiwan; yjhsu@csie.ntu.edu.tw
\*  Correspondence: wjl@cmlab.csie.ntu.edu.tw

**Abstract:** This paper presents a blockchain-based time bank system on the basis of the Hyperledger Fabric framework, which is one of the permissioned blockchain networks. Most of the services provided by existing Time Bank systems were recorded and conducted manually in the past; furthermore, jobs for matching services with receivers were managed by people. Running a time bank in this way will cost lots of time and human resources and, worse, it lacks security. This work designs and realizes a time bank system enabling all the service-related processes being executed and recorded on a blockchain. The matching between services' supply-and-demand tasks can directly be done through autonomous smart contracts. Building a time bank system on blockchain benefits the transaction of time credit which plays the role of digital currency on the system. In addition, the proposed time bank also retains a grading system, allowing its members to give each other a grade for reflecting their degrees of satisfaction about the results provided by the system. This grading system will incentivize the members to provide a better quality of service and adopt a nicer attitude for receiving a service, which may positively endorse the development of a worldwide time bank system.

**Keywords:** blockchain; Hyperledger; time bank; service; cryptography

## 1. Introduction

The concept of time-based currency was created decades ago. However, the term "time bank" was first raised by Edgar S. Cahn in 1992 [1], and he also promoted an important philosophy: "equal time, equal the value". Years later, in 1995, he founded the Time Dollar Institute which is known as the "TimeBanks USA" today. What is the meaning of time banking or a time bank system? In a nutshell, a time bank system is a reciprocity-based work trading system where time is valuable as a currency. In Cahn's book [2], he listed five core values of a time bank system which are: "asset, redefining work, reciprocity, social networks, and respect". These core values spread around the world and motivated people worldwide to establish various time bank systems in their own communities. In the outbreak era of influenza and viruses, especially COVID-19 these days, the necessity and importance of building mutual assistance within communities increased significantly. This fact motivates us to devote ourselves to build tools and/or systems for straightening out the chaotic situation we are facing. As shown in this work, with the aid of modern technology such as blockchain, a time bank system could manifest its value in building community, inclusion, volunteerism, and social assistance.

Typically, a time bank system is held by a time bank committee and built in a local community. Furthermore, the system is exclusive, whereby only people who are known to the committee can join the system. Agents of a time bank have to manage every legal member's request-or-supply for a service through phone calls and record them on notebooks, before manually matching up the demand and the

supply for services, all of which are complicated tasks. Moreover, an exhaustive service recording is also very time-consuming.

All the works need to be handled, within a local or domestic district, by time bank system employees. It would be a good idea to transfer the function of a time bank system to the internet so as to enable the participants to directly upload their service demands and supplies to the internet, even though mobile devices.

However, these time bank systems are centralized in nature. Although a centralized system benefits the system's efficiency a lot, the participants have to build their mutual trust by relying on a trusted third party (TTP), the time bank system's administrant. A recent technology called blockchain provides a decentralized platform that allows people to interact with each other without any mediator or central controller. In this work, a blockchain-based time bank system was designed and realized on the basis of the Hyperledger Fabric framework.

This work is organized as follows: Section 2 gives the reason for choosing Hyperledger Fabric as our system implementation backbone and reviews some related works on blockchain-based time bank systems. Section 3 offers function-level overviews of the proposed system. The involved functional modules are explained in Section 4. Section 5 presents the implementation details of the proposed system and reports some measured performances. Challenges that our system might encounter in real-world applications are firstly depicted in Section 6. Finally, the conclusion and our future research directions are presented.

## 2. Background and Related Work

### 2.1. Blockchain Technology

Blockchain was popularized since the introductory of Bitcoin [3], which is built on top of a public permissionless chain allowing everyone to join and exchange values through the virtual currency called Bitcoin. Blockchain can be viewed as a decentralized ledger allowing direct peer-to-peer information transferring. In other words, it consists of a series of data that are shared by clusters of computers but not located on a single server. All the on-ledger recorded data are open and transparent to every on-chain node (computers). On-chain members can initiate a transaction and create a block associated with the transaction-related information. The block will be verified by thousands of nodes on the network, and the verified block will be added to the end of the mainchain [3]. Thus, blockchain stands for a chain of data, and each block of data is signed and guarded by cryptographic functions to make all the data immutable and secure.

Bitcoin uses this technology mainly for monetizing transactions. Ever since Bitcoin was invented by its mysterious creator, Satoshi Nakamoto, blockchain technology steadily grew in popularity, with ever-growing use-cases. However, despite the rise in popularity of blockchain tech, some people still question whether or not *decentralized* blockchains are a good idea [4].

Later, in 2014, an innovative blockchain platform called Ethereum was proposed by Vitalik Buterin [5], and it provides a decentralized platform for software development. Ethereum introduced the mechanism of smart contract to allow any kind of business logic to be autonomously executed on blockchain, and it enabled participants to build both decentralized financial and non-financial applications. After the appearance of Ethereum, there was a drastic increase in the number of blockchain-based applications being developed, and blockchain is regarded as a new paradigm shift of information technology.

### 2.2. Blockchain in Social Business and Social Credit Systems

In Reference [6], the authors pointed out that a human being's true value lies in serving other people, and they suggested constructing trustworthy and safe communities based on a blockchain-enabled social credits system (BLESS) that rewards the residents who participate in socially beneficial activities. Since this is a position paper, the proposed BLESS system was conceptually designed on the basis

of the Ethereum blockchain, and there was no system implementation-related report. Although the application domain is different from time banking, References [7,8] investigated the application of blockchain technology to address some of the key challenges faced by the domain of social business (SB). Reference [7] tried bridging the gap between the potential usage of blockchain in social businesses by designing, developing, and evaluating an Ethereum blockchain-based crowdlending platform of social business. This work shows that blockchain enables otherwise unsustainable social business models, mainly by replacing intermediaries, and it requires changes in software engineering practices. Reference [8] modeled a small example of a micro-credit use-case from microfinance activities of SB using a semi-formal modeling approach. In addition to identifying that blockchain provides solutions that enhance trust, transparency, and auditability in SB activities, Reference [8] also identified challenges related to creating a native cryptocurrency for SB, as well as barriers to infrastructure and technology adoption by the different stakeholders in SB.

### 2.3. Blockchain in Regular Banking Industry

In facing the impact of economic transformation, internet development, and financial innovations, the regular banking industry requires urgent transformation and is seeking new growth avenues. As indicated in References [9–11], blockchain technology has the potential to facilitate global money remittance, smart contracts, automated banking ledgers, and digital assets; therefore, it could revolutionize the underlying technology of the payment clearing and credit information systems in banks. However, the regulation and actual implementation of a decentralized system are problems that remain to be resolved. Therefore, Reference [9] promoted the establishment of a "regulatory sandbox" and the development of industry standards, while Reference [10] addressed the key issues that must be considered in developing such ledger-based technologies in a banking context. Reference [11] reported, specifically, the application of blockchain in the China Foreign Exchange Trade System.

In a nutshell, blockchain technology benefits a time bank system in several ways. Firstly, blockchain enables peer-to-peer token trading. In a time bank system, there is a token called time credit and it is used for quantifying the value of services in terms of time units. All the on-chain data are immutable. That is, no one can modify the data recorded on the blockchain, such as the service-related data or the balance in the time credit wallet. The reward rate and the matching algorithm are transparent to everyone on the blockchain. The reward-related information is written on smart contracts. The reward rate is the rate for rewarding service providers, as detailed in Section 4. Paying transaction fees is essential in a few currently existing time bank systems. In a blockchain-based time bank system, the payment of transaction fees is optional.

### 2.4. The Hyperledger Fabric Blockchain

Hyperledger Fabric (Fabric for short) is one kind of permissioned blockchain aiming mainly at business applications [12]. It is a flexible open-source system that also allows the involvement of smart contracts, which are named "Chaincodes" in Fabric. Chaincodes can be programmed by using several conventional programming languages such as Golang and JavaScript unlike the smart contracts in Ethereum, which are restricted to a domain-specific language, Solidity. Other components in Fabric, such as the consensus protocol and the distributed database, are modularized and pluggable, which enable the system to be applicable to many practical applications.

Since Fabric is designated for deploying and operating on permissioned blockchains, it supports the Membership Service Provider (MSP) for providing identity checks to all nodes in the network. Anyone who wants to join to the network has to firstly enroll and get an identity. Furthermore, Fabric belongs to the group of consortium blockchains, in that it has a certificate authority (CA) in the network. In other words, the centralization of Fabric, as well as the proposed Time Bank system, would be higher than that of its Bitcoin Blockchain-based counterparts. Nevertheless, as mentioned in Section 1, a reliable time bank system is centralized in nature because its participants have to build their mutual trust by relying on a centralized TTP, the time bank system's administrant.

In Fabric, there is another particular structure called the channel. Channels separate all on-chain organizations into different groups, and one channel owns its attached ledger. That is, only the members belonging to the same channel can share the data on the specific ledger. This configuration fulfills the requirement of conducting confidential transactions between members. Members are in control of the visibility of data through channels, and this characteristic is significant and useful in lots of business applications.

In Fabric, the consensus is reached in the phase of ordering service so that consensus-reaching only needs the supports of endorsing peers. Therefore, Fabric can deal with many more transactions than Ethereum in the same period of time. Additionally, there is no built-in cryptocurrency in Fabric although the developers can still define and issue their own tokens in Fabric, such as the "time credit" in this study.

To sum up, for building a time bank system with blockchain technology, we decided to use Fabric rather than Ethereum due to the following considerations:

(1) Fabric has a higher rate of transactions per second (TPS) since reaching consensus in Fabric does not require agreement from all on-chain members. Thus, the finality of a Fabric transaction is immediate.

(2) Paying transaction fees in Fabric is not mandatory.

(3) There is no cost limit in a Fabrics Chaincode unlike the gas limit in an Ethereum-based smart contract.

(4) Member identification can be verified by the equipped MSP in Fabric.

(5) Fabric can use channels to partition members' supply-and-demand pairs according to their geographical locations; therefore, users can query the service data only recorded on a specific channel. That is, usually, users can focus only on nearby services. Noticeably, Fabric also enables cross-channel interaction if users need to exchange the service launched on another channel.

(6) The Fabric network is highly scalable. Once a new community is willing to join the time bank, the only work needed to be done is to create a new channel that includes those newcomers as the legal community members. In other words, the number of members and communities can be easily scaled up.

## 3. System Overview

### 3.1. Overview of the Application Scenarios

The time bank system presented in this work includes several different entities. The first one is the service receiver (SR), who upon needing help sends a service demand (or request) to the blockchain network. The second one is the service provider (SP) who is capable of providing some services within some specific time periods. SPs post when they are available and what kinds of services they can supply (provide) to the Blockchain. SRs and SPs are the majority members of a time bank, and there are a handful of special entities called grading managers. Grading managers are responsible for managing secret keys and using those keys to reveal some members' grades which were encrypted on the ledger.

When we say a service-exchange process is complete, it means that the fulfillment of a matched demand-and-supply service pair is accomplished. To complete a service-exchange, the entities of a time bank system have to follow the processing steps shown in Figure 1.

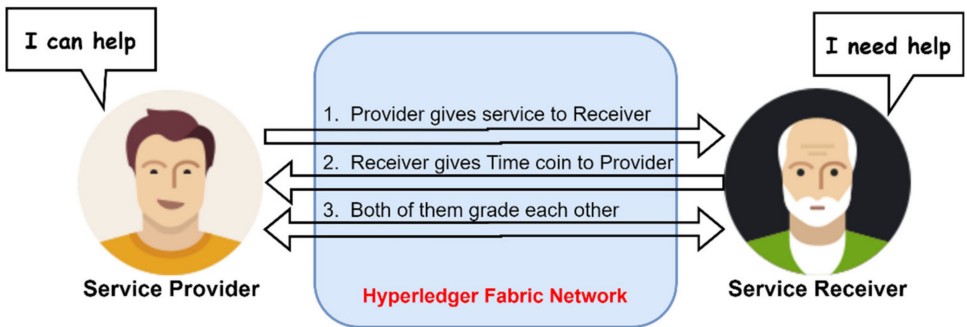

**Figure 1.** Basic processing steps of the proposed time bank system.

- Step 1: SRs and SPs post their service supply-and-demand proposals on the ledger. After a proposal is verified by the blockchain, the system tries to match it up with all other not-yet-matched service proposals already recorded on the ledger.
- Step 2: If a provider's supply matches a receiver's demand, that specific service supply-and-demand pair is said to be settled (or the service-exchange is paired for short). In other words, the deal of a service-exchange pair between an SP (say Alice) and an SR (say Bob) is made.
- Step 3: After their service-exchange is paired, Bob pays Alice some amount of pre-negotiated time credits (or time coins). The balance records in their on-ledge wallets will be updated accordingly.
- Step 4: In the end, both SP and SR grade each other. That is, Bob grades Alice according to the provided service quality; likewise, Alice grades Bob according to Bob's attitude while receiving the service. All the grades will be encrypted and recorded on the ledger.

*3.2. System Architecture Overview*

As shown in Figure 2, the proposed time bank system can be divided into multiple layers. Firstly, the user-layer consists of the enrolled SPs and SRs, who can access each client-site through an interface configured in the application-layer so as to connect to the blockchain. Notice that peers bridge between users and the blockchain network; therefore, a user can go through one of the selected peers to interact with the network for completing well-defined business logic functions by invoking specific Chaincodes.

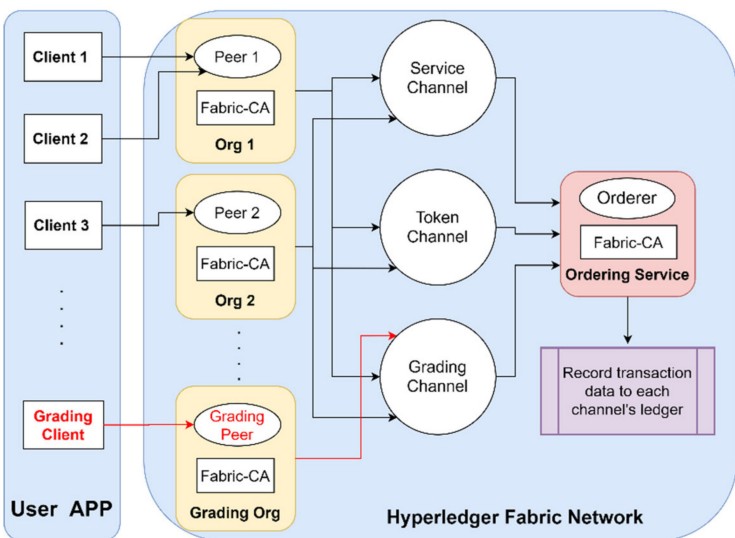

**Figure 2.** Architecture of our Fabric-based time bank system.

Users can not only put their data on blockchain but also request data from the blockchain-based ledgers. In addition, every organization has a specific entity, called fabric-CA, which is responsible for the registration of identities and the issuance of certificates.

The Fabric network in our system is composed of three different channels. The first is called the service channel (SC), which deals with all the service-related networking issues. The second is called the token channel (TC), which is responsible for collecting all wallet data from the ledger. The third is called the grading channel (GC), which allows the enrolled members to grade each other. Each channel is attached with Chaincodes running on peers and provides a platform for program execution to transfer transactions. In a Fabric framework, all the transactions are evaluated and reach consensus in the ordering phase. Finally, all the transactions are sent to a special entity called the "orderer". The orderer puts all the transactions in execution order and updates them on the ledgers, for both endorsing and non-endorsing peers, with respect to their original channels.

### 3.3. Specifications of the Channel Structure

In Fabric, only the members of a given channel have the rights to record their data or query existing data from the ledger associated with the channel; thus, the membership of each channel needs to be specified. All channels have the members Peer1, Peer2, . . . , etc., which represent different user clients. This means that users can access all channels to do different jobs such as posting services, transferring tokens, and grading services. The only difference is that the grading peer, which represents the client of the grade manager, belongs only to GC. In other words, the grade manager has no right to post services and/or access the wallets.

### 3.4. Preliminary Assumptions about our Time Bank System

In the real world, a time bank will encounter lots of challenges and constraints. Therefore, some assumptions and/or conditions must be specified before elaborating on each part of our prototype system. Basic assumptions about the design of the proposed time bank system include the following:

(1) The time credit, which is the token for quantifying the value of a service. Each unit of time credit (i.e., the time spent of service) is traded equally regardless of the substantiality of the services rendered. This assumption is to make the contributions of different services be valued equally so that there is no discrimination in the service offered.

(2) The inclusion of rewarding function makes the total time credit amount of our system increase slowly. The possible inflation of time credit is disregarded in this study.

(3) In a real-world time bank, it is necessary to ensure the safety of all members. Although the time bank committee validates the members' backgrounds in the enrollment stage, no one can guarantee that the enrolled members would always be law-abiding. However, for simplifying the design, we assume that every system participant would always be law-abiding and would not commit a crime.

(4) In our service-matching stage, the impact of geographical locations on the selection of service proposals is not considered. This assumption ignores the possible time spent on traveling in service offerings, i.e., we assume that all system participants live not too far away. If the location of the service is considered, the matching algorithm needs to calculate the distance and estimate the traveling time between the matched SP and SR; clearly, this will complicate the situation a little bit!

## 4. The Proposed Time Bank System

### 4.1. System Flow Chart

As described in Section 3, our time bank system consists of three different channels, and there are sequences of transactions executed in each channel. Figure 3 shows the information flow and processing procedures of the proposed time bank system. For ease of explanation, the operations of our system are divided into three phases, as explained in the subsections below.

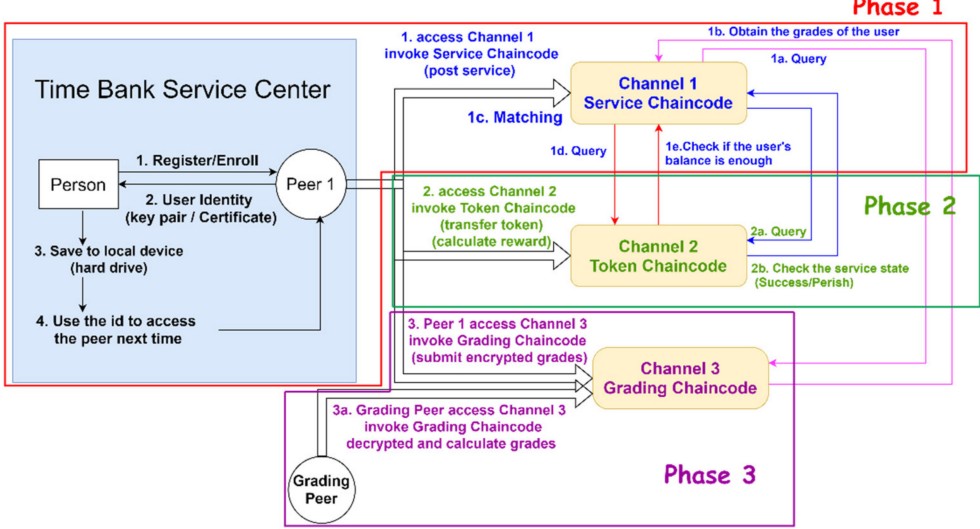

**Figure 3.** The information flow and processing procedures of the proposed time bank system.

*4.2. Phase 1: Service Posting and Matching*

Phase 1 includes the three stages of member enrollment, service posting, and service matching. In this phase, users always connect to SC and invoke service Chaincodes.

(1)  *Enrollment Processes:* At first, a user has to participate in the time bank system. He or she needs to enroll to become a valid system member since our time bank is a permissioned blockchain-based system. The enrollment process is depicted in the gray block of Figure 3.

A candidate user has to access one peer of the system and sends an enrollment request to the fabric-CA with his or her certificate, which contains the user's name and some personal related attributes. Then, fabric-CA (which usually represents the time bank committee) checks the applicant's name and all other related attributes on the certificate. Once the validation is a success, the fabric-CA will sign the certificate. Any certificate without fabric-CA's signature is viewed as invalid. The returned messages from fabric-CA include not only a signed certificate but also a private-and-public key pair. These keys are used for providing digital signatures to the on-chain blocks. Moreover, fabric-CA is equipped with its own database for recording all the names that were enrolled. This design prevents users from enrolling duplicated names in a time bank.

After an applicant is enrolled as a valid member, with the associated certificate and key pair, he or she can invoke a transaction to TC for creating his or her wallet, which is the place to save time credits (see Figure 4 for details). In our system, a valid member who creates a new wallet will get 10 free initial time credits. In order to promote the pre-described core values of time banks, this work tends to encourage participants to accept other services being offered. Getting free tokens without any service offering, in the beginning, can also attract people to join our time bank system. In GC, the enrollment of a new member will also initialize a zero grade for him or her.

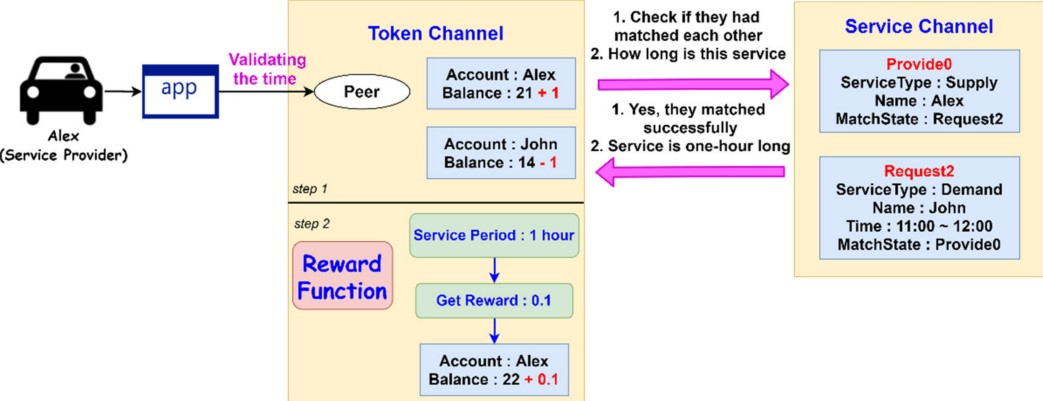

**Figure 4.** The information flow and the processes in the token channel (TC).

(2)  *Service Posting:*  On-chain enrolled members can now start posting their service-related data (i.e., service proposals) on the blockchain. They can use their key pairs to access the peer, upload their service proposals, and invoke the PostService function on service Chaincodes in SC. Users can access different peers which implies they could go to different time bank centers on the premise that these peers belong to the same channel. In our current design, it is assumed that the involved service attributes include service type, name, date, service class, location, and time period. These attributes are the basis for service matching.

(3)  *Service Matching:*  After the service proposals are posted on SC, the proposed system tries to find matches as much as possible. Without a loss of the generality, this paper just uses a simple (intuitive) matching scheme to find matches while there are other complicated matching algorithms [13–15], whose adoption would enhance the matching performance. Either SP or SR needs to take all the attributes (except the location-related one) into comparison so as to find the matched services. Figure 5 shows the adopted service0matching scheme in SC.

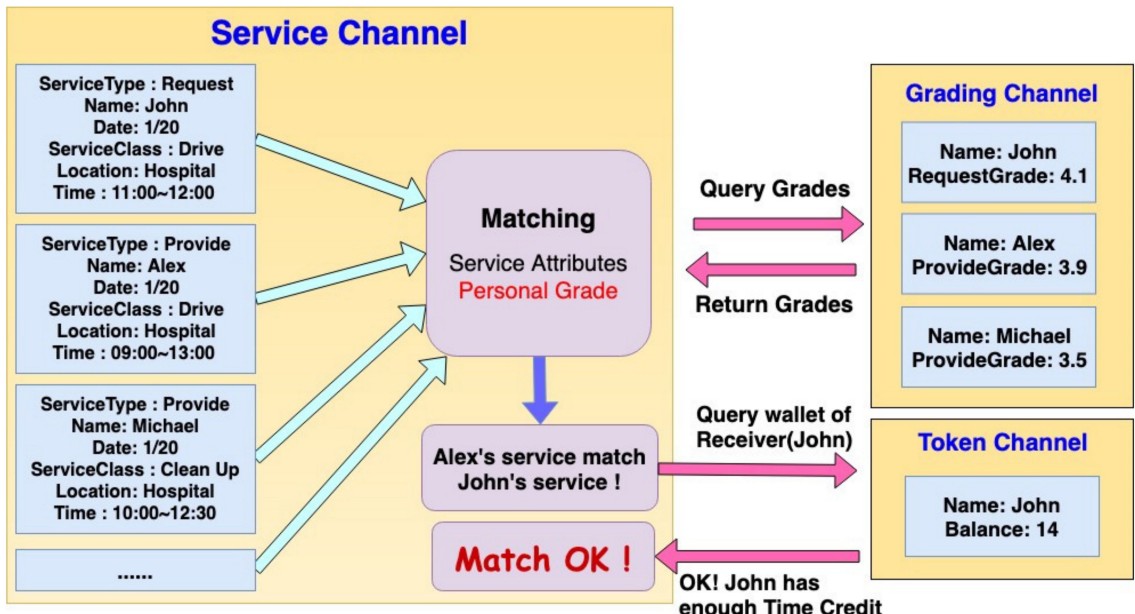

**Figure 5.** The service-matching process in the service channel (SC).

In addition, to take various attributes into account for finding a match, there is another important factor, i.e., the service grade involved in our matching process. As previously described, GC allows participants to grade each other after they exchanged services. The value of a grade roughly indicates

the quality of the service provided by an SP or the degree of temperateness to the service an SR receives. Those low-grade members should get punishment from the system so as to retain the time bank's core values. A possible solution to this is to make those low-grade members getting a match much harder. In other words, a lower grade denotes a lower priority and probability that he or she can get a match with others. For example, as shown in Figure 5, both Alex and Michael can provide the driving service for John; however, since Alex has a higher ProvideGrade, our matching scheme will pair up John's service to Alex's service rather than Michael's.

(4)  *The Interaction Constraint on Cross-Channel Chaincode:* As mentioned before, each channel owns its individual ledger in Fabric. In other words, different channels are built on different blockchains, i.e., they stay in different world states and record different ledger data. The applications in the Fabric-based system allow users to access the same peer, connect to different channels, invoke different Chaincodes, and search data from different ledgers. However, there is a strict constraint on the Chaincodes' cross-channel communications. In the official document of the Hyperledger Fabric, there is a paragraph mentioning this constraint. As the document states, access to Chaincodes in channels other than the channel targeted by the first Chaincode in the invokement will be read-only. Thus, in cross-channel communications, users from other channels have rights only to query an on-chain ledger but not to change a world state data.

Although the user service-related process is run in SC, the grade data are recorded on the ledger in GC, which must be considered in the matching step. To deal with this cross-channel referencing issue, a designated Fabric cross-channel communication function in SC is applied to query the grade data in GC. In the same way, the balance of the receiver's wallet can be queried while the wallet data are recorded in TC. Furthermore, the user in SC can check whether the receiver's balance is sufficient enough to pay the required provider's fee, once their services are matched.

Due to the previously described constraint on cross-channel interaction, users in the original channel cannot change the world state or upload data to another channel. That is, the time credit cannot be directly transferred from receiver to provider at the same time when their services are matched. Users have to go back to the application layer and connect to TC for executing the transactions regarding the time credit.

### 4.3. Phase 2: Token Transferring

This phase focuses mainly on token transferring, that is, transfer of the token between SR and SP after their services are exchanged.

*(1) Token Transferring:* The whole token transferring-related process in TC is shown in Figure 4. At first, a user executes an application to access TC and invokes the token transferring function embedded on token Chaincodes. In this case, after SR and SP exchange their services, SP wants to get the time credit for the service he or she provided. Due to the passivity of blockchain, a token will be transferred only when a user accesses it and invokes a smart contract on it. Since only SP has the motivation to get time coins, a token transfer must be activated by an SP. He or she can directly invoke the token transferring function in TC through an application and gets the time credit as a return. However, there are several rules to obey for preventing some possible fraud generated by SPs.

Firstly, the system needs to check whether the time an SP requests the token transfer is later than the time that the corresponding services were exchanged. This means that a token transferring function should not be activated until the corresponding service's exchange time passes. Otherwise, the system must check whether SP and SR matched their services exactly. Secondly, at the time an SP is executing the token transferring function in TC, the application-related function defined in token Chaincodes should already have successfully obtained the service-related data, which are recorded on SC. Finally, the response to users' queries about SC should indicate that both service states of SP and SR are successful, and then the token transferring function transfers the time credit from SR's wallet to SP's wallet.

*(2) Reward Function:* An important goal of the time bank is to promote the willingness of supplying services; thus, the system needs to incentivize the participants to provide services; therefore, a reward function is designed in our system. Once a service-exchange is completed, the reward function will automatically count the service time that SP provided. In our system, every hour of service-providing will gain 0.1 units of time credit as a reward. Of course, the above two numbers are adjustable parameters. The process of our reward function is also shown in Figure 4.

### 4.4. Phase 3: Grading System

*(1) An Overview:* Generally, a grading system is used to collect all grades associated with a specific member before showing the average grade to him or her. However, most of the participants in a time bank may be neighbors in the same community or live in geographic close-by regions; therefore, directly revealing every individual's grade will cause disharmony.

Hence, in our system, every single grade is hidden, and only the average grade is released. The period of average grade releasing is, of course, defined as another system parameter. For example, it is proper to reveal the average grade of an SP once five individual grades are received. When it comes to hiding a plaintext, the most common way is to use a cryptographic encryption scheme. However, our grading function needs to calculate the average of a certain amount of ciphertexts as a functional return. Fortunately, there is a system facing a similar problem to us, which is the E-voting system. Comparing to those E-voting systems [16–19], our grading system has a significant difference, that is, our private key must never be revealed. This is because an E-voting system is a one-time process such that all the keys will be re-generated if another run of the election is held. However, in our grading system, the key pair cannot be changed every time the average grade is calculated and revealed. In other words, the private key adopted in the grading system should never be released.

Due to the above-mentioned reasons, our grading system was designed to be a semi-decentralized system, that is, as outlined in Section 2.2, there is a TTP joined in this system but there are some restrictions on its functionality. This third-party entity is called the grade manager and is responsible for generating storing and concealing the encryption key. The grading system and the grade manager are detailed in the subsections below.

*(2) The Processes of Our Grading System:* The full mark is defined as 5.0 in the grading system. In addition, the grade is separated into two types, provide-grade and receive-grade, since people receive grades when they are either SP or SR.

In our grading system, users have to conduct the encryption process before submitting their grades to the ledger. That is, the encryption is done in an off-chain phase. As for the encryption process, the asymmetric encryption scheme is selected since the encryption and decryption are executed by different parties. In this work, the well-known asymmetric encryption scheme RSA [20] is adopted.

The steps of the grade submission and average grade calculation, as illustrated in Figure 6, are explained below.

(a)   Firstly, both users (SP and SR) encrypt their plain grades with *public-key* 1, and the encrypted results are respectively defined as *ciphertext-1a* and *ciphertext-1b*. Users have to keep these ciphertexts as their commitment for system checking later.

(b)   Then, the users encrypt *ciphertext-1a* and *ciphertext-1b* with *public-key* 2, and the results are denoted as *ciphertext-2a* and *ciphertext-2b*. Users submit *ciphertext-2a* and *ciphertext-2b* to the ledger so as to hide the real grade value.

(c)   The grade manager periodically accesses the grading peer to query those encrypted grade data such as *ciphertext-2a* and *ciphertext-2b*. Since the grade manager owns the *private-key 1* and *private-key 2*, they can firstly decrypt the *ciphertext-2a* back to *ciphertext-1a with private-key* 2 and then decrypt the result again with *private-key* 1 to get the plain grade data. When the grade manager collects five plain grade data of someone, they can calculate the average grade for that person.

(d)   Finally, the grade manager submits the average grade and the ciphertexts which are obtained
      by decrypting the encrypted data on the ledger with *private-key* 2. These ciphertexts, such as
      *ciphertext-* 1*a*, are regarded as the "commitments" of the grade data decryption. Once a user finds
      out that his average grade was revealed on the ledger, he can query the commitments to check if
      it is the same ciphertext as the one he kept, for example, *ciphertext-*1*a*. Notice that the *private-key*
      1 is always publicly available on the ledger so that every user can query those commitments
      and decrypt them with *private-key* 1 to re-calculate and verify the integrity of the average grade.
      This mechanism can prevent the grade manager from submitting modified average grades.

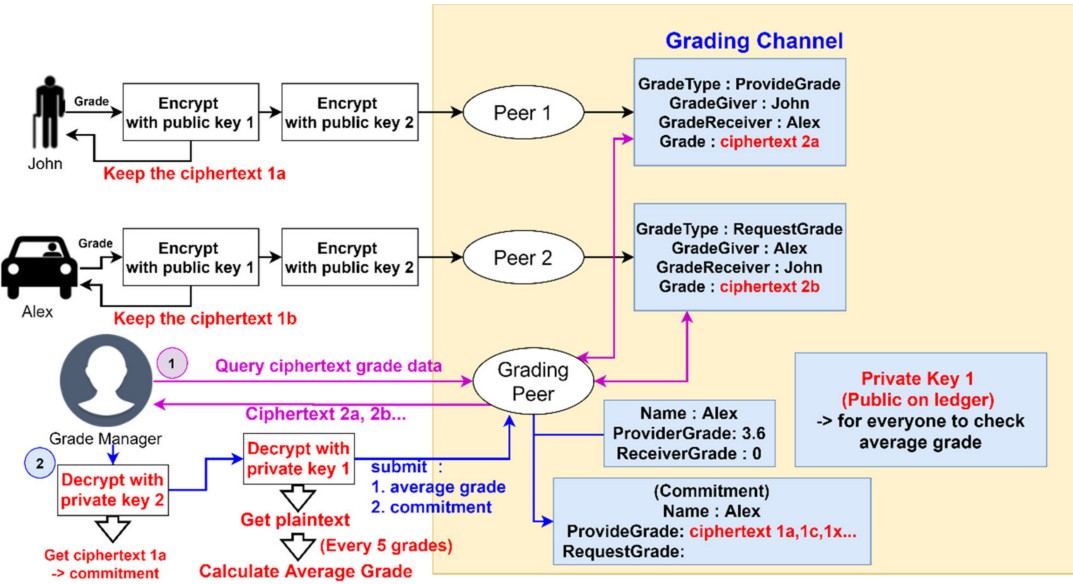

**Figure 6.** The information flow and the processes in the grading channel (GC).

As mentioned earlier, it is required to calculate a member's average grade after every five grades
are received. Thus, every time the grade manager calculates the average grade for someone, they are
supposed to submit the average grade and the five "commitments", which indicates that the grade
manager did obtain five encrypted grade data and did calculate their average. That is, as a member
provides five services to others and receives five grades, which are classified as a provide-grade by
associated SRs, he or she can get his or her provide-grade revealed.

In the steps of grade submission and average grade calculation, those two pairs of public-key
and private-key are generated by the grade manager at the initial stage. The two public keys are
transmitted to the local application site for doing off-chain encryption. The first private key (*private-key*
1) is submitted to the ledger for the users to decrypt the received commitments for checking the average
grade. The second private key (*private-key* 2) is kept by the grade manager and is only accessible to the
grade manager. The grade manager can query the blockchain network periodically so as to decrypt the
grade data and calculate the average grade for the member who receives enough grades. In real-world
applications, this grade manager can be a member of the time bank committee and he or she may get
paid for adequately fulfilling this grade calculation job.

Notice that the RSA encryption scheme adopted in this system includes a padding function for
adding extra randomness such that the result of RSA is not deterministic anymore. This padding
function fulfills the specification of the Public Key Cryptography Standards (PKCS)#1 version 1.5.
Therefore, encrypting the same grade data twice will produce two different ciphertexts in our system.
Let us look at the grading process, as shown in Figure 6. No one can reproduce the *ciphertext-*2*a* or
*ciphertext-*2*b* by trying to re-encrypt all possible grades with *public-key* 1 and *public-key* 2. Similarly,
re-encrypting those commitments with *public-key* 2 will also not generate the same results as that of

the encrypted grade. For example, re-encrypting the *ciphertext*-1*a*, which is the commitment recorded on the ledger, will not result in *ciphertext*-2*a*. Therefore, although any commitment in the ledger can be decrypted to reveal the corresponding real grade value, our system never discloses to users who submitted this grade.

## 5. Experimental Results

### 5.1. Environmental Settings

The complete operations of the proposed time bank system are schematically illustrated in Figure 7. The numbers listed in the left column indicate the sequences of phases in our system. The blocks in yellow denote the processes related to users, and the blocks in blue denote the processes related to the grade manager. Notice that all our Fabric network is built on a personal computer (PC) with an Intel 7700K central processing unit (CPU) and 8 GB of random-access memory (RAM), running Ubuntu 16.04 (64 bits). The adopted Fabric version is 1.4, and all the nodes run in Docker where the version is 18.09.3. All the Chaincodes and the Fabric network components are written in golang language where the version of golang is 1.10.8. The applications are written in node.js8.15.9 with npm 6.9.0.

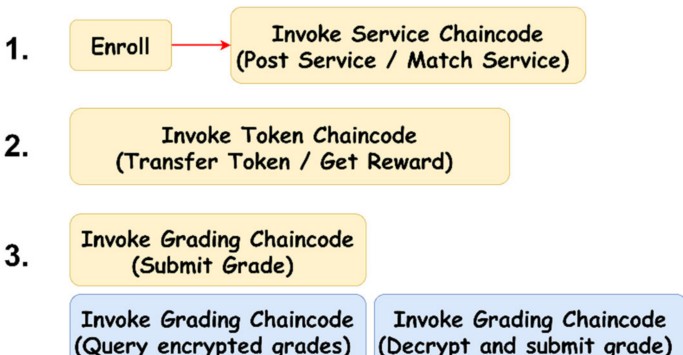

**Figure 7.** The simplified block diagram of overall operations in the proposed time bank system.

### 5.2. Time Bank-Related Operations

In phase 1, every user has to begin with the enrollment process. If a new identity whose name is "Cody" completes his enrollment, then there is a key pair and a document containing the attributes of this identity. After enrolling, Cody can use an application to input all the related service states and post them on the blockchain.

Next is the execution of the matching processes. After Cody posts his service, our system compares it with all existed services and shows all matchable ones. Then, the system queries the grades of the service owners of these candidates and chooses the highest-grade candidate before matching. As mentioned earlier, when a service request is submitted, the system has to check the sufficiency of SR's balance which is recorded on the TC. Finally, the service gets a successful match. A measurable performance index is the total execution time of the whole process from when the service is posted to when the service is matched, which is about 10 to11 seconds in our current system implementation. Of course, this time spent depends heavily on the number and scale of the involved communities and the efficiency of the adopted matching algorithm.

In Phase 2, SP invokes token Chaincodes to get the time credit for his or her services and automatically activates the reward function. The wallet on the state contains two attributes: account and balance, where account is used to record users' names and balance records the time credit that the user has. If an SR is requesting a one-hour service, the SP will get 1.1 units of time credit where one is from SR and 0.1 is the reward. According to our simulation, the execution time of Phase 2 is about 5 s.

In Phase 3, both SP and SR submit their grades. Then, the grades are off-chain encrypted with RSA. At first, the two public and private key pairs of RSA are generated by another key managing program.

Then, the user can invoke grading Chaincodes to submit their grade to others. In the states of the grade, there are four statements which are "grade type", "grade giver", "grade receiver", and "grade", where grade type indicates the grade is given to SP or SR and "grade" shows the encrypted grade data. Since the grades were already encrypted with RSA, their ciphertexts are shown in base-64 encoding.

Then, the grade manager queries the ledger to get the encrypted grade. Notice that the grade manager also has to go through the enrolling process with MSP. The grade manager decrypts the encrypted grade data one time to get the commitments and two times to get the plain grade data. Once the grade manager collects five grades of someone, they can calculate that person's average grade and submit it to the ledger.

The time cost of encrypting the grade data two times with RSA is about 7.5 s in our system. This execution time is calculated because the user has to input the grade no later than when the encrypted grade is recorded on the ledger.

To show the response time more precisely, as suggested by one of the anonymous reviewers, we measured the overall time spent under the following experimental conditions: (1) there were three different kinds of possible services; (2) the number of concurrent users (i.e., number of SR + number of SP) was assumed to be less than or equal to 10; (3) the reported time spent was the average value obtained by repeating the experiment 10 times. Our experimental results of the average system response time versus the number of system users are depicted in Figure 8. We also found that the system response time increases as the number of possible services increases, but the effect is negligible (0.1 to 0.2 s increase even when the number of possible services is increased from three to 100).

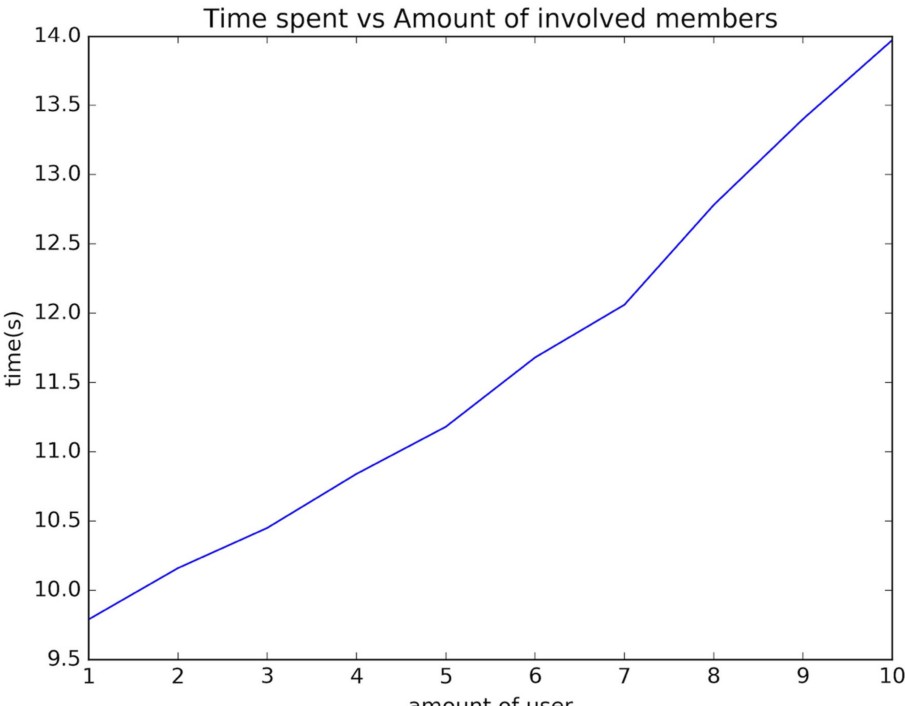

**Figure 8.** Our system response time versus the number of concurrent system users.

As pointed out by one of the anonymous reviewers, in practice, payment can be effective in the sub-second interval if the most updated user-oriented blockchain technology is used. This efficacy comes from the reduction in both the bank and the user in terms of the time and costs related to the transaction because several intermediaries could be removed from the process. Since the same merit holds for the proposed time bank system, according to our current realization (no optimization skill was applied), the estimated time credit payment, as shown in Figure 8, could be effective in the

sub-minute range. Of course, as previously described, this time spent depends heavily on the number and scale of the involved communities and the efficiency of the adopted matching algorithm.

## 6. Discussion and Conclusions

### 6.1. Challenges Encountered in Real-World Applications

Notice that posting a lot of services within a very short time period can be viewed as a possible attack on our system. That is, in service posting and matching processes, this can occur if there is an enrolled member who has no time credit in his or her wallet but posts a lot of service requests in a very short time period. Since the proposed system will continuously search for matches of the posted services with other not-yet-matched services, lots of interruptions will be caused. These heavy interruptions may flood all peers and disrupt the normal system operations. This phenomenon is treated as a possible attack similar to the well-known denial-of-service (DoS) attack. A possible solution to this attack is asking SRs to pre-pay a certain amount of the deposit at the moment when a service request is posted; if someone's balance is deducted to zero in his or her wallet, this person will be unable to post a service request. Of course, this way may cause some trouble to those people who highly rely on time bank systems and often post lots of urgent service requests.

There are lots of challenges if our time bank system is applied in the real world.

(1) The system needs to find an elegant mechanism to control the total amount of time credits. Due to the existence of the reward function, the total amount of time credits may increase as time goes by, which may lead to inflation.

(2) An effective incentive strategy is a must for attracting people to join the system.

(3) If system participants can provide only a few categories of services, the service exchange system can never operate in good condition, that is, quite a lot of service requests cannot be fulfilled.

(4) It is essential to consider whether the time credit can be transferred to fiat money. If yes, the exchange rate will be hard to determine.

(5) If the amounts of SPs and SRs are quite unbalanced, we need to determine how the system can keep operating normally.

(6) Of course, investments from governments and donations from charitable institutions will speed up the success of our system and resolve the cold-start problem the system will face.

### 6.2. Discussion

As pointed out by one of the anonymous reviewers, the idea of exploiting blockchains to implement grading and reputation systems is already available in the scientific literature [21–23]. For example, Reference [21] also provided a permissioned blockchain-based architecture and implementation of a system that allows the agents to interact with each other and enables tracking how their reputation changes after every interaction. In facing the application scenarios of the choice of reliable partners for cooperation with Internet of things (IoT) devices (or agents) migrating across different environments, another example was addressed in Reference [22], where most of their members are unreferenced, corresponding to their trustworthiness. If the trustiness of an IoT device is modeled according to the reputation capital of each agent, the effective solution to the problem of reputation management in multi-agent systems [21] can be applied to the problem considered in Reference [22]. Similarly, as a final example, a new reputation system for data credibility assessment in vehicular networks was proposed in Reference [23]. In this system, vehicles rate the received messages based on observations of traffic environments and pack these ratings into a "block". Each block is "chained" to the previous one by storing the hash value of the previous block. Then, a temporary center node is elected from vehicles, and it is responsible for broadcasting its rating block to others. Based on ratings stored in the blockchain, vehicles can calculate the reputation value of the message sender and then evaluate the credibility of the message.

### 6.3. Conclusions and Future Research Directions

From the above discussions, although the application domains are different from one and other, the idea of exploiting blockchains to implement grading and reputation was proven to be useful for building mutual trust among networked members and increasing the security of the whole system. The unique attribute of our system and the major difference from the above-mentioned works lies in the involvement of the double-encrypted process of the proposed grading system, which is an important design feature for maintaining the community's harmony in our work.

In summary, a time bank system designed based on the Hyperledger Fabric framework was presented in this work. With a time bank, people who are in need of help or able to provide help can post their service inquiries on the blockchain. Currently, our system will match up the posted services with a naïve matching method. Once people complete their service exchange, the one who provides the service can execute the token transferring function to get his or her time credit as a return. Then, both SR and SP can give each other a grade to reward or punish the other due to the provided service quality or the attitude in receiving services. In the future, we will move our research focus to solve the problems addressed in Section 6.1. Moreover, some additional functions may be added in order to fulfill the needs of social welfare. For example, a charity foundation function can allow members to donate their time credits and then transfer the donations to those who demand lots of help. Another direction is to build a function for providing extrinsic rewards such as the government's subsidies to those people in need. Of course, an efficient matching algorithm is worthy of exploration.

As described in Section 1, building mutual assistance systems during pandemics of influenza and other viruses is becoming even more important. Noticeably, the demand for building an effective and efficient mutual assistance system within local communities, like the one proposed in this work, will have long-term benefits for the harmony of mankind. Therefore, even though there are lots of challenges in front of us, if our current time bank system is to be applied to the real world, we will continue devoting our efforts to conquer the above-mentioned problems as much as possible, in the near future. It is our humble hope that, one day, this system can be demonstrated to be useful to people in need and to promote harmony in communities around the world.

**Author Contributions:** Conceptualization, Y.-T.L. and J.-J.L.; methodology, Y.-T.L.; software, Y.-T.L.; validation, Y.-T.L. and J.-J.L.; formal analysis, Y.-T.L. and J.-J.L.; investigation, J.-J.L., J.Y.-J.H., and J.-L.W.; data curation, Y.-T.L.; writing—original draft preparation, Y.-T.L.; writing—review and editing, J.-L.W.; visualization, Y.-T.L.; supervision, J.Y.-J.H. and J.-L.W.; project administration, J.-L.W.; funding acquisition, J.-L.W. All authors read and agreed to the published version of the manuscript.

**Funding:** This research was funded by the Ministry of Science and Technology, Taiwan, under the grant number MOST 108-2218-e-002-055.

**Conflicts of Interest:** The authors declare no conflict of interest.

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
