# Peer review of "A Time Bank System Design on the Basis of Hyperledger Fabric Blockchain"

_futureinternet, doi:10.3390/fi12050084_

Round 1

Reviewer 1 Report

A Time Bank System Design on the Basis of Hyperledger  Fabric Blockchain

The authors propose a Blockchain-based time bank system based on the Hyperledger Fabric framework. The system allows all related service processes to be run and logged on blockchain. In addition, the Time Bank proposal also maintains a rating system that allows its members to rate themselves by reflecting their degree of satisfaction with the results provided by the system.This is an interesting approach. Below are my comments to the authors:

  1. The introduction does not clearly show the motivation of the work. Give it more depth.
  2. The authors can complement their study on Blockchain technology, in the reference cited in:

Valdivia, L. J., Del-Valle-Soto, C., Rodriguez, J., & Alcaraz, M. (2019). Decentralization: The failed promise of cryptocurrencies. IT Professional, 21(2), 33-40.

  1. A payment can be effective in seconds using an optimized user-oriented blockchain systems. Which will reduce both the bank and the user the time and costs related to the transaction. This is because several intermediaries would be removed from the process. Better explain the assumptions about our Time Bank System.
  2. Could you improve the explanation of the proposed solution about the centralization? Is the any other solution?
  3. While it may be outside the intent of the paper, explaining ways that the potential for centralization can be mitigated would be an interesting and useful addition to the paper. Cryptocurrency is rapidly evolving and expanding, but it is constrained by the extent to which people trust it. The trust engendered by using blockchain depends partially on it remaining decentralized. So research into ways that decentralization can be compromised and mitigated are both relevant and very important. There is other research into the potential for centralization of cryptocurrency blockchains. For example, discuss how control of Bitcoin is centralized by limited set of entities. Exploring centralization is another aspect that should be further explored.
  4. The discussion is poor and the authors could delve into the interruptions.

Author Response

Point to point reply to reviewer-1’s comments for the paper entitled: A Time Bank System Design on theBasis of Hyperledger Fabric Blockchain, co-authored by Yu-Tse Lee, Jhan-Jia Lin and Jane Yung-Jen Hsu, and Ja-Ling Wu

Manuscript ID: futureinternet-785744| Type of manuscript: Article

Manuscript ID: futureinternet-785744

Reviewer-1’s comment-1: The introduction does not clearly show the motivation of the work. Give it more depth.

Our-reply: As suggested, a few sentences are added at the end of the first paragraph of Section 1, to explain why the addressed subject is important, especially nowadays, which clearly shows our motivation for carrying out this work.

Reviewer-1’s comment-2: The authors can complement their study on Blockchain technology, in the reference cited in:

Valdivia, L. J., Del-Valle-Soto, C., Rodriguez, J., & Alcaraz, M. (2019). Decentralization: The failed promise of cryptocurrencies. IT Professional, 21(2), 33-40.

Our-reply: Thanks to the reviewer for bringing the above write-up to our attention, which is now listed as the fourth reference in the revision. As suggested, a paragraph is added at the end of the first paragraph of Section 2.1 to reflect the concerns about the capability of applying decentralized blockchains to cryptocurrencies.

Reviewer-1’s comment-3: A payment can be effective in seconds using an optimized user-oriented blockchain systems. Which will reduce both the bank and the user the time and costs related to the transaction. This is because several intermediaries would be removed from the process. Better explain the assumptions about our Time Bank System.

      Our-reply: Thanks to the reviewer for reminding us about the major merit of applying blockchain technology to build Time Bank systems: The removal of a certain number of intermediaries from the process which makes the reduction in both the bank and the user the time and costs related to a transaction (a service-exchange). In response to this insightful comment, a new paragraph is added at the end of Section 5 to report the estimated time spent on getting a payment effective, in the proposed system. The reported estimate is made according to our conducted simulation results, currently.

Reviewer-1’s comment-4: Could you improve the explanation of the proposed solution about the centralization? Is the any other solution?

       Our Reply: This is another insightful comment given by the reviewer, thanks again! The reason why we chose Hyperledger Fabric, a permissioned blockchain, as the basic platform to realized our Time Bank system is: Building mutual trust among registered members is a must to the success of a Time Bank system; however, building mutual trust in a fully distributed environment is a very challenging problem, even if distributed ledger technology (such as Blockchain) is applied. To solve this conflict, some restrictions must be placed on those centralized modules, such as CA and Grading Manager, in our system. In response to this valuable comment and emphasize the semi-decentralized characteristic of the proposed Time Bank system, several sentences are added at the end of the second paragraph of Section 2.2.

Of course, with the aid of complicated cryptographic mechanisms as we did in an E-voting work published in [16], the requirement of centralization could be released somewhat. We will not elaborate on this issue further since it seems out of the focus of this work.

Reviewer-1’s comment-5: While it may be outside the intent of the paper, explaining ways that the potential for centralization can be mitigated would be an interesting and useful addition to the paper. Cryptocurrency is rapidly evolving and expanding, but it is constrained by the extent to which people trust it. The trust engendered by using blockchain depends partially on it remaining decentralized. So research into ways that decentralization can be compromised and mitigated are both relevant and very important. There is other research into the potential for centralization of cryptocurrency blockchains. For example, discuss how control of Bitcoin is centralized by limited set of entities. Exploring centralization is another aspect that should be further explored.

       Our Reply: Our experiences with the issue of centralization and/or decentralization obtained from developing Blockchain-based applications, especially in non-financial categories, such as E-voting, online Education, online auction, health-record sharing, and Time Banking, do coincide with reviewer’s points mentioned in the above comment. That is, research into ways to compromise and mitigate the decentralization characteristic of blockchain is interesting and of great importance. For example, as addressed in our reply to comment-4, to ensure mutual trust on a blockchain-based E-voting system [16], the assistance of some rather complicated cryptographic techniques – ring signature and time-lock encryption scheme is a must to maintain the system’s centralization behavior. Actually, a similar situation occurred in our development of a blockchain-based Online Teacher-Student Interaction platform, to appear in [A]. However, as pointed out by the reviewer, the subject seems outside the intent of this paper, we did not elaborate on this issue further in the revision.

[A] Cheng-Ting Tsai, Ja-Ling Wu, and Yu-Tzu Lin, “A Blockchain-Based Fair and Transparent Teacher-Student Interaction Platform," accepted by "ICALT 2020: International Conference on Advanced Learning Technologies" will be held on Oct 29-30, 2020 in Paris, France. 

Reviewer-1’s comment-6: The discussion is poor and the authors could delve into the interruptions.

       Our Reply: To eliminate the shortage, we have restructured our manuscript by merging the original Sections 6 and 7 into the new Section 6 of the revision, where sub-section 6.1 reveals the potential challenges we might face, if the proposed Time Bank system is used in practice. Sub-section 6.2 discusses the advantages or differences between some newly added related works and our proposal. While sub-section 6.3 presents the conclusion and possible future research directions of this work.  Moreover, a new paragraph is added at the end of sub-section 6.3 of the revision to emphasize the importance and possible contributions of this write-up.

Reviewer 2 Report

The paper presents an architecture to implement a Time Bank System using permissioned blockchains. Specifically, the authors uses Hyperledger Fabric, an open-source permissioned blockchain implementation, and its components to manage:
- the "supply-and-demand" process between Service Receivers and Service Providers;
- the transfers and the wallets with the currency of the Time Bank, i.e. the Time Coins;
- a grade-based evaluation of the services requested/offered by the users of the Time Bank, in order to allow users to build their own reputation in the platform.

Despite the idea is well described and the English language of the manuscript is good and clear, the paper suffers of three major issues.

1. The paper does not include enough elements to evaluate its contributions to the state of the art. There is a section called "Background and Related Works", but only background on the blockchain technology and Hyperledger Fabric is presented. In such section, there are no references to the scientific literature and no comparisons of the presented work with related research, therefore the reader is not able to evaluate how the proposed research advances the state of the art.

To fix this problem, the authors might consider to extended the introduction and the background and related works sections. In the introduction they can write a paragraph explicitly stating which contributions their work adds to the state of the art of blockchain applications for banks, service exchanges, social credits, and/or in general. In the background and related works section, the authors might consider to write an entire subsection dedicated to the related works. Here, the authors should write a brief literature survey citing other papers to compare their work with other relevant research in their domain. For example:
- in [1] a blockchain-based architecture for social credit is described. In the related works subsection the authors should briefly summarize such work, and explain which advantages their proposal has w.r.t. to [1].
- Similarly, in [2,3] blockchain-based designs for social business are presented. I understand the application domain is different from Time Banking, but it would be interesting for the reader understanding the differences and the advantages of the presented research w.r.t. similar works in other application domains.
- In addition, what are the differences of the authors' proposal w.r.t. regular bank applications of blockchain technologies (see, for example, [4,5,6])? Are there requirements in common between money transfers and time banking which might be relevant for a blockchain application? Are there requirements, instead, which make blockchain applications for regular banking not applicable to Time Banking?
- Moreover, the idea of exploiting blockchains to implement grading and reputation systems is already available in the scientific literature (see, for example, [7,8,9]). Which are the advantages or differences between the cited works and the research presented by the author of this research on Time Banking, in the use of blockchains to build (also) a reputation system?

These are examples of what can be discussed in a "Related Works" subsection. The authors might consider even do add other works they found which might be related to their research.

2. Strictly related to my previous point, there is a possible lack of novelty. Of course, as the authors present an application, I was not expecting a technical contribution to blockchain technology and I'm not asking it. However, I'm concerned that the application itself is similar to other blockchain-based works for social credits or regular banking, as those I cited in my previous point. Therefore, the authors should explicitly write in their paper:
- which features of their design make it actually new or better than similar applications;
- try to add more details to explain why their proposal is particularly suited for Time Banking;
- try to speculate about the possible application of their design to other domains.
Furthermore, the following sentences

"agents of a Time Bank have to manage every legal member’s request-or-supply for a service through phone calls and record them on notebooks, and then match up the demand and the supply for services manually, which is a very complicated task. Moreover, the service recording is also a very time-consuming job."

looks very strong to me. I'm not expert at all about time banks, but are they completely managed manually, via phone call and notebooks, as it appears from such sentences? What I understand from such sentences is that no software and no dbms are used to manage information in time banks. Instead, I would expect to have a certain degree of automation for processing information, as it happens in banks. Therefore, I think the authors should clarify this point.

3. The experimental results presented in section 5 are not accurate enough. For example:
- the authors do not write the number of service receivers and service providers in the system when the experiment are executed. I expect that such numbers might influence the response time of the system, from a posted service to a matched service. I also expect that, when there are many providers for the same service, the total execution time to match that service will be higher;
- the authors do not clarify the number of repetitions of the experiments to obtain the values they presents. For example, the total execution time is said to be between 10 and 11 seconds. In how many experiments did the authors obtain such numbers?
- the given results are a little bit too vague. Instead of writing that the total execution time is about 10/11 seconds, the phase 2 time is about 5 seconds, and the time of encrypting is about 7.5 seconds, the authors should give a precise number, which will be the average obtained executed "N" tests, along with its standard deviation. In my opinion, that would be way more informative than the given numbers.
- a graph showing the trends of such times depending on the number of service receivers and providers, i.e. the agents of the system, would at least give an idea of the scalability of the system, which would add a great value to the paper.

To summarize, in my opinion the listed three major issues should be fixed before considering the paper worth of publication.

Minor comments:

- there are some small formatting problems along the paper. For example:
-- in line 104 and 105 different fonts are mixed;
-- in line 242 the (4) is not in italized as the previous points;
-- in line 282 the (2) is not in italized as the previous points;
-- in lines 421-430 the numbering of the enumerated list are not coherent (in general, fonts of enumerated list along the paper look bigger than they should be);

- figures look stretched and/or at low resolution. The authors should use vectorial images or, at least, images with a higher resolution.

References

[1] Xu, R., Lin, X., Dong, Q., & Chen, Y. (2018). Constructing trustworthy and safe communities on a blockchain-enabled social credits system. In Proceedings of the 15th EAI International Conference on Mobile and Ubiquitous Systems: Computing, Networking and Services (pp. 449-453).

[2] Schweizer, A., Schlatt, V., Urbach, N., & Fridgen, G. (2017, December). Unchaining Social Businesses-Blockchain as the Basic Technology of a Crowdlending Platform. In ICIS.

[3] Mukkamala, R. R., Vatrapu, R., Ray, P. K., Sengupta, G., & Halder, S. (2018, December). Converging blockchain and social business for socio-economic development. In 2018 IEEE International Conference on Big Data (Big Data) (pp. 3039-3048). IEEE.

[4] Guo, Y., & Liang, C. (2016). Blockchain application and outlook in the banking industry. Financial Innovation, 2(1), 24.

[5] Peters, G. W., & Panayi, E. (2016). Understanding modern banking ledgers through blockchain technologies: Future of transaction processing and smart contracts on the internet of money. In Banking beyond banks and money (pp. 239-278). Springer, Cham.

[6] Wu, T., & Liang, X. (2017). Exploration and practice of inter-bank application based on blockchain. In 2017 12th International Conference on Computer Science and Education (ICCSE) (pp. 219-224). IEEE.

[7] Calvaresi, D., Mattioli, V., Dubovitskaya, A., Dragoni, A. F., & Schumacher, M. (2018). Reputation management in multi-agent systems using permissioned blockchain technology. In 2018 IEEE/WIC/ACM International Conference on Web Intelligence (WI) (pp. 719-725). IEEE.

[8] Fortino, G., Messina, F., Rosaci, D., & Sarne, G. M. (2019). Using blockchain in a reputation-based model for grouping agents in the internet of things. IEEE Transactions on Engineering Management.

[9] Yang, Z., Zheng, K., Yang, K., & Leung, V. C. (2017). A blockchain-based reputation system for data credibility assessment in vehicular networks. In 2017 IEEE 28th annual international symposium on personal, indoor, and mobile radio communications (PIMRC) (pp. 1-5). IEEE.

Author Response

Point to point reply to reviewer-2’s comments for the paper entitled: A Time Bank System Design on theBasis of Hyperledger Fabric Blockchain, co-authored by Yu-Tse Lee, Jhan-Jia Lin and Jane Yung-Jen Hsu, and Ja-Ling Wu

Manuscript ID: futureinternet-785744| Type of manuscript: Article

Since Reviewer-2’s first comment is rather long in length, for the ease of replying, we divide it into 3 sub-comments (a, b, c) and reply to them, respectively.

Reviewer-2’s comment-1: The paper does not include enough elements to evaluate its contributions to the state of the art. There is a section called "Background and Related Works", but only background on the blockchain technology and Hyperledger Fabric is presented. In such section, there are no references to the scientific literature and no comparisons of the presented work with related research, therefore the reader is not able to evaluate how the proposed research advances the state of the art.

Comment 1-(a)To fix this problem, the authors might consider to extended the introduction and the background and related works sections. In the introduction they can write a paragraph explicitly stating which contributions their work adds to the state of the art of blockchain applications for banks, service exchanges, social credits, and/or in general. In the background and related works section, the authors might consider to write an entire subsection dedicated to the related works. Here, the authors should write a brief literature survey citing other papers to compare their work with other relevant research in their domain. For example:
- in [B1] a blockchain-based architecture for social credit is described. In the related works subsection, the authors should briefly summarize such work, and explain which advantages their proposal has w.r.t. to [B1].
- Similarly, in [B2, B3] blockchain-based designs for social business are presented. I understand the application domain is different from Time Banking, but it would be interesting for the reader understanding the differences and the advantages of the presented research w.r.t. similar works in other application domains.

Our Reply to comment 1-(a): Thanks to reviewer-2 for bringing references [B1, B2, B3] to our attention, as suggested a new sub-section (Sub-section 2.2), entitled “Blockchain in Social Business and Social Credit Systems” is added in Section 2 of the revision. From the reviewer’s informative comments, we learned that blockchain does enable otherwise unsustainable social business models, which removes intermediaries and requires changes in software engineering practices.

Comment 1-(b): In addition, what are the differences of the authors' proposal w.r.t. regular bank applications of blockchain technologies (see, for example, [B4, B5, B6])? Are there requirements in common between money transfers and time banking which might be relevant for a blockchain application? Are there requirements, instead, which make blockchain applications for regular banking not applicable to Time Banking?

Our Reply to comment 1-(b): Thanks to reviewer-2 for bringing references [B4, B5, B6] to our attention, as suggested a new sub-section (Sub-section 2.3), entitled “Blockchain in Regular Bank Industry” is added in Section 2 of the revision.

Instead of comparing the Similarity-and-Difference of the roles the Blockchain might play in Time Bank and a regular Bank, we would like to point out the differences between Bank Credit and Mutual-Trust Credit, as shown in Table 1.

Technology-wise, Blockchain is applicable to both Time Banking and regular Banks. To us, the differences between a TimeBank and a regular Bank come from the goals of providing services, the customer bases, the related administration regulations, and the characteristics of ‘credits’ they are concerned with. With the popularity and modularity of Pure-Software Bank, we do believe the technology gap (including Blockchain) between a TimeBank and a regular Bank is narrowing as time goes by.

 Comment 1-(c): Moreover, the idea of exploiting blockchains to implement grading and reputation systems is already available in the scientific literature (see, for example, [B7, B8, B9]). Which are the advantages or differences between the cited works and the research presented by the author of this research on Time Banking, in the use of blockchains to build (also) a reputation system?

These are examples of what can be discussed in a "Related Works" subsection. The authors might consider even do add other works they found which might be related to their research.

Our Reply to comment 1-(c): This part is the most insightful comment about our work, thanks to reviewer-2 again for reminding us that the idea of exploiting blockchains to implement grading and reputation systems has already been presented in different application domains; therefore, we should address the advantages or differences between the cited works and our proposal in a bit more detail. In response to this comment, an overview of the cited works is added at the beginning of the Discussion sub-section (sub-section 6.2 of the revision) and a new paragraph for depicting the major difference between the cited works and our research is added in the beginning of sub-section 6.3, as part of the conclusion of this write-up.

Since Reviewer-2’s second comment is rather long in length, for the ease of replying, we divide it into 3 sub-comments (a, b) and reply to them, respectively.

Comment 2-(a): Strictly related to my previous point, there is a possible lack of novelty. Of course, as the authors present an application, I was not expecting a technical contribution to blockchain technology and I'm not asking it. However, I'm concerned that the application itself is similar to other blockchain-based works for social credits or regular banking, as those I cited in my previous point. Therefore, the authors should explicitly write in their paper:
- which features of their design make it actually new or better than similar applications;
- try to add more details to explain why their proposal is particularly suited for Time Banking;
- try to speculate about the possible application of their design to other domains.

     Our Reply to comment 2-(a): As compared with the cited references, our work has the following strong points, and therefore, makes it actually a new contribution to Time Bank systems.

  1. Our work is realized on the Hyperledger Fabric, which is a more suitable blockchain platform for Time Bank than the Ethereum counterpart. We presented both the design and the realization results in detail, not just the design concept as shown in some of the cited works. As suggested by another reviewer, we have presented the estimated time spent on completing a payment (service-exchange) of our system, in the revision.
  2. The double-encryption mechanism of our Grading system is very useful for keeping the harmony of the involved members, which is our unique design feature and an advantage over all existing works related to Time Bank systems. As mentioned in the revision, it is our belief that the ability to keep Harmony of a Community is of specific importance in the outbreaking era of influenza and viruses, especially the COVID-19 in these days.
  3. According to our work on conducting Blockchain-based applications, the semi-decentralized characteristics and the requirement for securing sensitivity on-chain data (we are faced in this work) are common in E-voting, Online Grading, and online auction systems, we do think some of our design experiences can be applied to other applications, directly.

Comment 2-(b): Furthermore, the following sentences

"agents of a Time Bank have to manage every legal member’s request-or-supply for a service through phone calls and record them on notebooks, and then match up the demand and the supply for services manually, which is a very complicated task. Moreover, the service recording is also a very time-consuming job."

looks very strong to me. I'm not expert at all about time banks, but are they completely managed manually, via phone call and notebooks, as it appears from such sentences? What I understand from such sentences is that no software and no dbms are used to manage information in time banks. Instead, I would expect to have a certain degree of automation for processing information, as it happens in banks. Therefore, I think the authors should clarify this point.

    Our Reply to comment 2-(b): Good Comment! Let us give a short brief about the current status of existing Time Banking systems (in reality, most of them are called the Community Exchange Systems (CES), instead), first. Up to 2018, CES is comprised of more than 1000 exchanges in about 100 countries, making it the first global network of alternative exchange systems. While it is a global network, it is fully decentralized with each exchange operating independently of the others and having its own management team and rules of operation. Today’s CES comprises of multiple small local exchanges instead of a global one, in which each community has its own rules and administers its own exchange. Moreover, each community has its own currency and unit of value (usually based on national currency or time). Each community has a conversion rate relative to other currencies, where the conversion rate is derived from local hourly rates. Since Timebanking is a kind of Non-Money (Noney) Exchange service, as mentioned in sub-section 6.1, there are lots of challenges to be met for making it widely accepted by the real world. Actually, lacking of investments from governments and donations from charitable institutions make Time Banking lies still on the cold-start state in most of the countries. This explains why most of CES are still managed manually. Of course, this situation has been largely improved for some developed countries, such as TimeBank USA; however, no software and no DBMS are used to manage the information in time banks is a common practice in the real world. This fact is one of the motivations for us to devote ourselves to do this work.

Since Reviewer-2’s third comment is rather long in length, for the ease of replying, we divide it into 3 sub-comments (a, b, c, d, e) and reply to them, respectively.

Comment 3-(a): The experimental results presented in section 5 are not accurate enough. For example: the authors do not write the number of service receivers and service providers in the system when the experiment is executed. I expect that such numbers might influence the response time of the system, from a posted service to a matched service. I also expect that, when there are many providers for the same service, the total execution time to match that service will be higher;

Our Reply to comment 3-(a): Thanks to reviewer-2 for giving this valuable comment, which polished our presentation of experimental results a lot. To increase the accuracy of our experimental results, we take the numbers of online SR and SP into account when we calculate the response time of the system in the revision.

We do agree the reviewer’s remark that the response time is proportional to the number of SPs of the same service, this is because we picking out the SP with the highest service grade, if the current matching algorithm is used, whose time spent do proportional to the number of providers for the same service.

Comment 3-(b): The authors do not clarify the number of repetitions of the experiments to obtain the values they present. For example, the total execution time is said to be between 10 and 11 seconds. In how many experiments did the authors obtain such numbers?

     Our reply to comment 3-(b): Yes, the numbers of online SR and SP do influence the response time of our system. Actually, we did address this fact in the second paragraph of subsection 5.2 in the original paper and at the end of subsection 5.2 of the revision. As suggested, we report the estimated system response time by repeating the experiment 10 times.

Comment 3-(c): The given results are a little bit too vague. Instead of writing that the total execution time is about 10/11 seconds, the phase 2 time is about 5 seconds, and the time of encrypting is about 7.5 seconds, the authors should give a precise number, which will be the average obtained executed "N" tests, along with its standard deviation. In my opinion, that would be way more informative than the given numbers.

Our reply to comment 3-(c): Totally agree with the reviewer’s remark, as indicated in the previous reply, the parameter N= 10, in our experiments.

Comment 3-(d): A graph showing the trends of such times depending on the number of service receivers and providers, i.e. the agents of the system, would at least give an idea of the scalability of the system, which would add a great value to the paper.

Our reply to comment 3-(d): As suggested, the graph of “the response time vs. the number of agents” (Figure 8) is presented in the revision.

Comment 3-(e): To summarize, in my opinion the listed three major issues should be fixed before considering the paper worth of publication.

Our reply to comment 3-(e): The authors would like to thank reviewer 2 for giving the above-mentioned valuable comment, which does polish the presentation of this work a lot!

Comment 4: Minor comments:

- there are some small formatting problems along the paper. For example:
-- in line 104 and 105 different fonts are mixed;
-- in line 242 the (4) is not in italized as the previous points;
-- in line 282 the (2) is not in italized as the previous points;
-- in lines 421-430 the numbering of the enumerated list are not coherent (in general, fonts of enumerated list along the paper look bigger than they should be);

- figures look stretched and/or at low resolution. The authors should use vectorial images or, at least, images with a higher resolution.

     Our reply to comment 4: Thanks, we have corrected the above-mentioned formatting problems as much as we can in the revision.

As suggested, we have re-drawn all the images again and tried to make them as clear as possible in the revision!

Comment 5: References

[B1] Xu, R., Lin, X., Dong, Q., & Chen, Y. (2018). Constructing trustworthy and safe communities on a blockchain-enabled social credits system. In Proceedings of the 15th EAI International Conference on Mobile and Ubiquitous Systems: Computing, Networking and Services (pp. 449-453).

[B2] Schweizer, A., Schlatt, V., Urbach, N., & Fridgen, G. (2017, December). Unchaining Social Businesses-Blockchain as the Basic Technology of a Crowdlending Platform. In ICIS.

[B3] Mukkamala, R. R., Vatrapu, R., Ray, P. K., Sengupta, G., & Halder, S. (2018, December). Converging blockchain and social business for socio-economic development. In 2018 IEEE International Conference on Big Data (Big Data) (pp. 3039-3048). IEEE.

[B4] Guo, Y., & Liang, C. (2016). Blockchain application and outlook in the banking industry. Financial Innovation, 2(1), 24.

[B5] Peters, G. W., & Panayi, E. (2016). Understanding modern banking ledgers through blockchain technologies: Future of transaction processing and smart contracts on the internet of money. In Banking beyond banks and money (pp. 239-278). Springer, Cham.

[B6] Wu, T., & Liang, X. (2017). Exploration and practice of inter-bank application based on blockchain. In 2017 12th International Conference on Computer Science and Education (ICCSE) (pp. 219-224). IEEE.

[B7] Calvaresi, D., Mattioli, V., Dubovitskaya, A., Dragoni, A. F., & Schumacher, M. (2018). Reputation management in multi-agent systems using permissioned blockchain technology. In 2018 IEEE/WIC/ACM International Conference on Web Intelligence (WI) (pp. 719-725). IEEE.

[B8] Fortino, G., Messina, F., Rosaci, D., & Sarne, G. M. (2019). Using blockchain in a reputation-based model for grouping agents in the internet of things. IEEE Transactions on Engineering Management.

[B9] Yang, Z., Zheng, K., Yang, K., & Leung, V. C. (2017). A blockchain-based reputation system for data credibility assessment in vehicular networks. In 2017 IEEE 28th annual international symposium on personal, indoor, and mobile radio communications (PIMRC) (pp. 1-5). IEEE.

Our reply to comment 5: All the suggested references are included in the revision.

Round 2

Reviewer 1 Report

I agree with the present manuscript. The changes were made correctly.

Author Response

Thank you for your valuable and insightful comments, which polished our work a lot!

Reviewer 2 Report

The authors addressed all the comments of the reviewers and significantly improved the paper. I particularly appreciated that, in the paper additions, they decided to mention that those clarifications where asked by reviewers. That was not necessary or due, but the authors have been very honest in doing that.

As the paper has been significantly improved, I suggest acceptance in present form.

Author Response

Thank you for your valuable and insightful comments, which polished our work a lot!

Based on your suggestion, we have re-drawn nearly all figures and Tables, which do enhance the quality of our presentation significantly.

Thanks again!